# Introduction of Vector-Borne Infections in Europe: Emerging and Re-Emerging Viral Pathogens with Potential Impact on One Health

**DOI:** 10.3390/pathogens14010063

**Published:** 2025-01-12

**Authors:** Jacopo Logiudice, Maria Alberti, Andrea Ciccarone, Benedetta Rossi, Giorgio Tiecco, Maria Antonia De Francesco, Eugenia Quiros-Roldan

**Affiliations:** 1Department of Clinical and Experimental Sciences, Unit of Infectious and Tropical Diseases, University of Brescia, ASST Spedali Civili di Brescia, 25123 Brescia, Italy; j.logiudice@studenti.unibs.it (J.L.); m.alberti035@studenti.unibs.it (M.A.); a.ciccarone@unibs.it (A.C.); g.tiecco@unibs.it (G.T.); eugeniaquiros@yahoo.it (E.Q.-R.); 2Unit of Infectious and Tropical Diseases, ASST Spedali Civili di Brescia, 25123 Brescia, Italy; benedetta.rossi19@gmail.com; 3Department of Experimental Medicine and Public Health, School of Advanced Studies, University of Camerino, 62032 Camerino, Italy; 4Section of Microbiology, Department of Molecular and Translational Medicine, University of Brescia, ASST Spedali Civili, 25123 Brescia, Italy

**Keywords:** vector-borne, mosquitoes, tick, One Health, climate

## Abstract

The rise and resurgence of vector-borne diseases (VBDs) in Europe pose an expanding public health challenge, exacerbated by climate change, globalization, and ecological disruptions. Both arthropod-borne viruses (arboviruses) transmitted by ticks such as Crimean–Congo hemorrhagic fever and arboviruses transmitted by mosquitoes like dengue, Chikungunya, Zika, and Japanese encephalitis have broadened their distribution due to rising temperatures, changes in rainfall, and increased human mobility. By emphasizing the importance of interconnected human, animal, and environmental health, integrated One Health strategies are crucial in addressing this complex issue. Europe faces increased risk due to the expanding habitats of disease-carrying organisms, the spread of new species like *Aedes albopictus* since 2013, and increased movement of infected individuals between countries, leading European countries to implement strategies such as enhanced surveillance systems, public awareness campaigns, and prompt outbreak response strategies. However, the lack of both targeted antiviral therapies and vaccines for many arboviruses, together with undetected or asymptomatic cases, hamper containment efforts. Therefore, it is important to have integrated strategies that combine climate modeling, disease surveillance, and public health interventions to address expected changes in disease patterns due to global changes. This review explores the spread of arboviruses in Europe, highlighting their historical context, current transmission dynamics, and their impact on public health.

## 1. Introduction

Climate change plays a pivotal role as a major risk factor for the emergence and re-emergence of human infections. According to the International Panel on Climate Change (IPCC), predicted climate scenarios include frequent heatwaves, heavy rainfall, and ecological droughts [1,2]. These conditions have geographically expanded the range of permissive vectors, therefore increasing the potential spread of a variety of infectious diseases, including vector-borne diseases (VBDs), and among them, the so-called arboviruses (arthropod-borne viruses). VBDs require vectors to establish themselves in ecological niches favorable to their reproduction. Climate change and human impact on biodiversity lead to important changes in the circulation of vectors that find favorable conditions to establish new ecological niches in territories considered non-endemic [3,4]. Many of these diseases have historically circulated in tropical and subtropical areas. One of the best-known climate changes taking place in Europe is the tropicalization of temperate climates leading to the presence of tropical animal species in other macro-climatic areas [5,6]. The latest report by the European Environment Agency (EEA) expressed concern about the steady rise in temperatures in Europe [7]. The summer of 2024 was the hottest ever recorded in Europe with an average temperature 1.54° higher than the 1991–2020 average [8]. Pathogens thrive within specific temperature ranges, and higher temperatures accelerate their development within vectors. For instance, in the last 10 years, milder climates have facilitated the migration of *Aedes* mosquitoes (the main vectors of dengue, Zika, and Chikungunya viruses) northwards causing the *A. albopictus* mosquito to become established in 13 countries and 337 European regions [9,10]. Although evidence of a relationship between climate change, weather patterns, and vector disease transmission is growing, future predictive models need to be standardized, especially in relation to climate variability in different regions and the complexity of transmission modes [11].

The re-emergence of VBDs is the result of a complex interaction among various determinants including biological characteristics of viruses, the transmission cycle and interaction with animal reservoirs, climatic and microclimatic changes, and human influence on ecosystems [12]. For example, animal reservoirs, such as mammals, birds, and reptiles, act as natural reservoirs for pathogens, and their movements allow the spread of diseases. For example, bird migration has had an important impact on the autochthonous transmission of WNV in the wetlands of Europe [13].

This represents an urgent health concern that needs an integrated “One Health” approach [14]. Combined with a highly mobile human population traveling to and from high-risk regions outside Europe for business, entertainment, politics, and war, this can favor the dissemination of VBDs and increase the risk of new pandemics [15].

Through an integrated surveillance system, European countries have adopted relevant strategies in the fight against VBDs, mobilizing multiple sectors and disciplines aimed to sustainably balance the health of ecosystems, animals, and humans. For example, the European Union has several vector monitoring programs run by the Food Safety Authority (EFSA), including VectorNet, which aims at European-wide integrated monitoring of VBDs [16]. The management of VBD outbreak cases requires a multidisciplinary and integrated approach across medical and health disciplines.

The aim of this review is to summarize which viral infections have been documented in humans or in animals transmitted by vectors already established in Europe or by vectors introduced in Europe following climate change impacts on ecosystems, together with the evaluation of the arboviral infections transmitted by vectors still not present in Europe, responsible only for imported infections (Table 1).

Therefore, our review has the purpose of focusing only on arbovirus epidemics in European countries and it has not considered a broad range of arboviruses that are already endemic in Europe, such as the West Nile virus and Usutu virus, even if they have a high public health impact. Co-circulation of the two viruses, in fact, has been reported in many European countries leading to enhanced seasonal surveillance for early detection of these viruses both in humans and animals.

## 2. Search Strategy and Selection Criteria

We searched PubMed and Google Scholar for peer-reviewed, English-language quantitative studies published up to 7 November 2024 that examined the epidemiological and clinical features of emerging arboviruses listed by the WHO publication of 26 September 2024 [17]. Specifically, only emerging viral diseases transmitted by vectors including dengue virus (DENV), Chikungunya virus (CHIKV), Zika virus (ZIKV), yellow fever virus (YFV), Japanese encephalitis virus (JEV), O’nyong’nyong virus (ONNV), Rift Valley virus (RVF) (all transmitted by mosquitoes), Crimean–Congo hemorrhagic fever (CCHF) (transmitted by ticks), and Oropouche fever (OROV) (transmitted by *Culicoides* flies) were covered.

We excluded West Nile, Usutu, and tick-borne encephalitis as they are not emerging viruses but are currently endemic in Europe [18,19].

To better stratify the risk of introduction or the outbreak potential of each disease, we have grouped these arboviruses based on the epidemiological context in Europe, particularly considering the presence of vectors and documented cases in humans and/or animals (both autochthonous and imported), as follows:(1)Pathologies with active transmission and autochthonous cases in human OR in animals transmitted by mosquitoes permanently present in Europe (*Aedes albopictus* and *Culex* spp.):
-Chikungunya virus (CHIKV)-Dengue virus (DENV)-Zika virus (ZIKV)-Japanese encephalitis virus (JEV)
(2)Pathologies with active transmission and autochthonous cases in human OR in animals transmitted by ticks permanently present in Europe (*Hyalomma* spp.):
-Crimean–Congo Hemorragic virus (CCHV)
(3)Pathologies without autochthonous transmission but with the presence of competent vectors in Europe:
-O’nyong’nyong virus (ONNV)-Rift Valley fever virus (RVFV)-Yellow fever virus (YFV)
(4)Pathologies without autochthonous transmission and without the presence of the vector on European territory:
-Oropouche virus (OROV)

References for this review were identified through searches in MEDLINE and PubMed, using the following combination of terms: “Chikungunya Fever” OR “Zika Virus Fever” OR “Yellow Fever” OR “Japanese Encephalitis Virus” OR “O’nyong’nyong Virus” OR “Crimean-Congo Hemorrhagic Fever” OR “Oropouche Fever” AND “etymology” OR “epidemiology” OR “transmission” OR “clinical manifestations” OR “diagnosis” OR “treatment” OR “prophylaxis”.

Articles from governmental infectious disease surveillance bodies at the European level (European Centre for Disease Prevention and Control, ECDC) and national level were analyzed for the latest available epidemiological data as of 07 November 2024, as well as for recommendations and entomological studies on vectors [20].

## 3. Pathologies with Active Transmission and Autochthonous Cases in Human or in Animals Transmitted by Vectors Permanently Present Across Europe

Vector-borne diseases (VBDs) transmitted by competent vectors, such as *Aedes albopictus* and *Culex* species, which are permanently established in Europe, represent a growing public health challenge on the continent. These mosquitoes, widely distributed across Europe (Figure 1), are capable of transmitting a variety of arboviruses to humans and animals. The occurrence of pathologies with active transmission and autochthonous cases highlights the complex interplay among environmental conditions, vector biology, and viral adaptation. Understanding the distribution and dynamics of these vectors, along with the epidemiological context of the diseases they spread, is crucial for effective risk assessment and the implementation of targeted prevention and control measures.

### 3.1. Chikungunya Virus

Chikungunya virus (CHIKV) is an alphavirus of the *Togaviridae* family, transmitted to humans by the bite of female *Aedes* mosquitoes [21]. The clinical triad of CHIKV infection, typical of the acute phase, is characterized by rash, fever, and joint pain, but disease severity varies widely, ranging from a mild self-limited illness to severe neurological manifestations and long-term debilitating arthralgias [22].

In Africa, the primary CHIKV reservoir is represented by non-human primates (NHPs) and other vertebrates, like rodents and bats. It circulates in an enzootic cycle involving zoophilic *Aedes* spp. mosquitoes (*Ae. furcifer*, *Ae. taylori*, *Ae. africanus*, and *Ae. luteocephalus*) [23]. In humans, *Ae. aegypti* represents the primary vector for CHIKV, associated with urban or peri-urban transmission, although, over the past decades, *Ae. albopictus*, originally a zoophilic forest-dwelling mosquito species from Asia, has become an increasingly significant vector for CHIKV [24]. *Ae. albopictus* has a wider geographic distribution than *Ae. aegypt i*, including temperate regions, enabling virus transmission outside endemic areas. Although CHIKV is primarily transmitted through mosquito bites, maternal–fetal transmission has also been reported, which can occur intrapartum leading to high rates of infant morbidity [25].

After its initial isolation occurred in 1952 in Tanzania, it caused only sporadic outbreaks in Africa and Asia. In India, outbreaks were recorded in 1963 in Calcutta, followed by epidemics in Chennai, Pondicherry, and Vellore in 1964; Visakhapatnam, Rajmundry, and Kakinada in 1965; Nagpur in 1965; and Barsi in 1973 [26]. In the African continent, the first significant outbreaks occurred in Kenya and in the Comoros Islands (located on the southeastern coast of Africa) in 2004 [27].

Phylogenetic analysis identified three distinct lineages corresponding to their respective geographical origin: West African, East Central South African (ECSA), and Asian lineage. During Kenya’s outbreaks, an ECSA CHIKV strain emerged and led to the first large outbreak of CHIKV in 2004–2005, which was reported in the Indian Ocean Island of La Réunion and resulted in the appearance of a fourth phylogenetic lineage termed the Indian Ocean lineage [28]. During this outbreak, a significant number of travelers from high-income countries became infected, leading to the first recorded dispersal of CHIKV to other countries: in 2007, CHIKV was identified in Italy, with 205 confirmed cases reported between 4 July and 27 September 2007 [29]. Subsequent outbreaks were recorded in Malaysia in 2008–2009 and Bangladesh in 2008 [30]. In early December 2013, an autochthonous transmission of CHIKV was recorded in the Americas, starting on the Caribbean island of Saint Martin and expanding through the Caribbean during the first half of 2014 due to population mobility between the islands [31].

Since then, additional outbreaks have been reported in Mediterranean Europe: in 2010, southeastern France faced the concomitant emergences of dengue virus (DENV) and CHIKV, with only two autochthonous infections confirmed for each virus [16]. In 2014, an outbreak of 12 autochthonous CHIKV cases was reported in Montpellier (Southern France) [17]. In 2017, a significant outbreak occurred in Italy, concentrated in three main foci in the Lazio region (Anzio, Rome, and Latina), resulting in 200 confirmed cases and 202 probable cases out of 699 suspected autochthonous cases [12].

To date, treatment of patients with CHIKV is mainly based on the use of anti-inflammatory drugs for symptomatic relief.

Although vaccination strategies against CHIKV are varied, only a few vaccine candidates are under clinical trial evaluation. Currently, IXCHIQ, a live attenuated vaccine, is the only vaccine licensed for use in adults aged 18 years and older at increased risk of CHIKV infection [32,33].

### 3.2. Dengue Virus

Dengue is an arboviral disease transmitted by mosquitoes of the genus *Aedes*, endemic to tropical and subtropical regions, and periodically responsible for large outbreaks. Dengue virus (DENV) belongs to the *Flaviviridae* family and the *Flavivirus* genus. It comprises four serotypes (DENV-1, DENV-2, DENV-3, and DENV-4), each of which includes multiple genotypes [34]. Dengue fever presents with clinical symptoms in less than 20% of cases, manifesting as headache, fever, rash, and musculoskeletal pain, while dengue hemorrhagic fever (DHF) involves high fever, hepatomegaly, hemorrhagic symptoms, shock, and cardiovascular complications [35].

*Ae. aegypti* is the primary vector for dengue transmission, although *Ae. albopictus* also sustains transmission in rural and suburban areas. *Ae. aegypti* generally has a shorter extrinsic incubation period (EIP) for dengue virus, but *Ae. albopictus* still poses a substantial risk for local transmission in Europe, where this species is prevalent, and it has significant transmission competence [36].

Additional species, such as *Ae. furcifer*, *Ae. taylori*, and *Ae. luteocephalus*, are implicated in sylvatic transmission cycles, often involving NHP and humans in West African forested regions [37]. Although the vector competence of *Ae. aegypti* varies significantly among African populations due to factors such as viral serotype, environmental conditions, and geographic origin, it remains the primary dengue vector. In contrast, *Ae. albopictus* exhibits lower yet more consistent competence, contributing to dengue transmission in areas where it is prevalent [38].

In Europe, *Ae. aegypti* has largely been absent since the 1950s due to synthetic insecticide DDT use and vector control measures [39], while *Ae. albopictus* is now established across much of the continent. In Europe, *Ae. aegypti* has re-established only in Cyprus, along the eastern Black Sea coast, and in the outermost region of Madeira [40].

Dengue is the most widespread arboviral infection globally, attracting significant attention from health organizations, including the World Health Organization (WHO), which designated it as one of the “Ten Threats to Global Health” in 2019 [20]. The first recorded dengue epidemic occurred in 1779 in Jakarta, Indonesia, and Cairo, Egypt [41], although retrospective analyses suggest its presence in the Americas as early as the 17th century [42].

Although dengue is not endemic in the WHO European Region, sporadic autochthonous cases and outbreaks have been reported since 2010 in Croatia, France, Italy, Portugal (Madeira), and Spain, principally due to the genotype 1 [43].

In Europe, cases of local transmission have increased, with more than 300 autochthonous cases reported in 2024, including 213 (82 during 2023) in Italy, 85 (45 during 2023) in France, and 8 (3 during 2023) in Spain. No death was reported among autochthonous cases [44]. In Italy, as of 7 November, there were 213 autochthonous cases, with a median age of 45 years, a 50% male distribution, and no deaths, of which at least 179 cases (84%) were DENV2 [45].

Although current dengue cases in Europe are seasonal, experts suggest it is unlikely to become endemic without year-round transmission among mosquitoes. However, climate patterns and global travel may intensify localized outbreaks. Global control strategies, such as Wolbachia-infected mosquitoes, offer potential solutions but face challenges with *Ae. Albopictus* [46]. Moreover, the management of VBD outbreak cases requires a multidisciplinary and integrated approach across medical and health disciplines. One example was the management of the outbreak that occurred between August and October 2024 with 199 cases of autochthonous dengue virus transmission reported in the city of Fano, Marche. This outbreak required substantial resources for microbiological surveillance, patient care, and vector control [47].

Climate change is expected to expand the geographical range of dengue vectors, increasing the likelihood of outbreaks in temperate regions [48]. As of early 2024, over 13 million dengue cases and more than 8500 deaths were reported globally [49].

Current vaccination strategies against dengue mainly include two tetravalent vaccines: CYD-TDV (Dengvaxia) and TAK-003 (Qdenga). CYD-TDV, developed by Sanofi, is only approved for individuals who have already contracted dengue, as in HIV-negative individuals, it may increase the risk of severe forms of the disease due to the ADE (Antibody-Dependent Enhancement) phenomenon [50]. It confers a higher protection against the DENV-2 serotype (95.1%) than against other serotypes [51]. In contrast, TAK-003 demonstrated long-term efficacy and safety against all four DENV serotypes in previously exposed individuals and against DENV-1 and DENV-2 in dengue-naïve individuals [52].

### 3.3. Zika Virus

The Zika virus (ZIKV), classified within the genus *Flavivirus* and family *Flaviviridae*, is an arbovirus primarily transmitted by *Aedes* mosquitoes.

Most ZIKV infections are asymptomatic (75–80%). When symptomatic, the infection is mild. The most common clinical manifestations, in order of frequency, are itchy rash, low-grade fever, headache, arthralgia and myalgia, conjunctivitis, and diarrhea [53,54].

The Zika virus is primarily transmitted through the bite of infected mosquitoes from the *Aedes* genus, even if it can also be transmitted through blood transfusions and organ transplants and sexually. Different species of *Aedes* mosquitoes have been identified as potential carriers including *Ae*. *africanus*, *Ae. albopictus*, *Ae. hensilli*, and *Ae. aegypti*. However, *Ae. aegypti* and *Ae. albopictus* are the primary vectors. *Aedes* mosquitoes are diurnal, which means they often bite during the daytime, unlike many other species [55]. *Ae. aegypti* is particularly efficient in spreading the Zika virus, dengue virus, Chikungunya virus, and yellow fever virus in urban areas due to its preference for human habitats [56]. This type of mosquito is found in all tropical regions of the world. Some research suggests that global warming could encourage a further expansion of *Ae. aegypti* in Europe, North America, and other regions previously unsuitable for its survival. Due to its ability to adapt to urban environments and withstand adverse conditions, *Ae. aegypti* has become one of the most challenging mosquito species to control [57]

*Ae. albopictus*, or the Asian tiger mosquito, can also transmit Zika, even if it is less competent than *Ae. aegypti* in tropical environments [58]. Native to tropical and subtropical regions of Southeast Asia, *Ae. albopictus* has successfully adapted to temperate areas, with its current range extending to North America and Europe [59]. The role of *Ae. albopictus* as an efficient vector in Zika virus transmission has remained uncertain. It was first identified as a potential ZIKV vector during an outbreak in Gabon in 2007. However, the arrival of ZIKV in Europe provided further evidence that *Ae. albopictus* is indeed capable of supporting ZIKV transmission [60].

First identified in 1947 in a Macaca monkey in the Zika Forest of Uganda, the virus remained relatively unnoticed for decades because the ancestral African lineage of ZIKV was mainly confined to enzootic cycles between non-human primates [61]. The earliest documented human case was identified in Nigeria in 1952, marking the virus’s entry into human populations [62]. Around this period, serological evidence in Indonesia (1977–1978) revealed that 3.1% of febrile patients were positive for Zika, indicating the virus’s spread beyond Africa to Southeast Asia [63]. These findings underscore Zika’s gradual geographic expansion and rising human exposure over time. Nonetheless, few human cases with mild, self-limiting symptoms were documented before 2007, underscoring the virus’s relatively low impact before its resurgence.

After May 2007, the Zika virus emerged as a significant human pathogen, leading to outbreaks in the Pacific Islands, beginning with Yap Island in the Federated States of Micronesia and later spreading rapidly across French Polynesia and some islands of the Pacific [64].

In early 2015, the virus was diagnosed for the first time in the northeastern Brazilian states of Bahia and Rio Grande do Norte [65,66]. Since then, it has expanded northward through South and Central America, reaching Mexico by late November 2015. Moreover, in 2016, some autochthonous cases were identified in Florida and elsewhere in the U.S.A. [67]. In February 2016, the Zika virus was declared to be a public health emergency of worldwide importance by the World Health Organization (WHO) [68]. Until recently, Europe was considered a low-risk region for Zika transmission, primarily due to its climate, which is less favorable for the *Aedes* mosquito. However, in August 2019, a local case of vector-borne transmission was reported in the Mediterranean coastal department of Var (city of Hyeres), within an area of permanent *Ae. albopictus* presence. This event marked the first known case of autochthonous transmission in Europe, raising concerns about the virus’s potential to establish itself even in temperate regions [69].

Currently, there is no specific therapy for Zika virus infection. New treatments, and also existing drugs, have been evaluated for their ability to counteract the virus through different mechanisms.

Given the lack of an effective vaccine, preventive measures, especially regarding mosquito control and sexual transmission, are crucial in controlling the spread of Zika. The WHO recommends that travelers going to endemic areas for the Zika virus should take measures to prevent mosquito bites (using insect repellents, wearing long-sleeved clothing, and sleeping under mosquito nets), avoid travel during pregnancy, if possible, and use condoms or avoid unprotected sexual contact. These precautions are particularly important for pregnant women, who are at higher risk of complications from Zika virus infection [70].

### 3.4. Japanese Encephalitis Virus

Japanese encephalitis virus (JEV) is a virus belonging to the family *Flaviviridae* and genus *Flavivirus* that was first isolated in the brain of a patient who died of encephalitis in Japan in 1924 [71]. JEV infection is often asymptomatic. Less than 1% of cases develop clinical symptoms. In endemic areas, symptomatic and severe forms predominantly affect young adults and children. The mortality rate among patients with encephalitis is high, estimated to be between 20% and 30%. Those who survive severe JEV infection may suffer from long-term neurological complications, such as cognitive impairment, behavioral changes, motor deficits, and recurrent seizures [72].

JEV is a zoonotic, vector-borne virus, mainly transmitted by *Culex* mosquitoes. *Ardeidae* waterbirds, especially egrets (*Egretta garzetta*) and herons (*Nycticorax nycticorax*), are natural reservoirs, and pigs, both domestic and feral, serve as amplifying hosts [73,74,75].

*Culex* mosquitoes are active during the evening and night and prefer rural or peri-urban areas with stagnant water. These mosquitoes feed mainly on cows (a dead-end host for JEV) and pigs (an amplifying host) that act as maintenance hosts in endemic regions. The proximity of pig populations is the main risk factor for JEV transmission to humans [76]. Other domesticated animals (such as cattle, horses, buffalo, and goats) and wild animals (like marsupials) are susceptible to JEV infection but are considered dead-end hosts. This means that they may become infected but develop low viremia, making transmission to mosquitoes impossible [77]. The promiscuity between wild and domestic animals and the presence of intensive animal farms are at the root of some VBD transmission [78]. Humans are considered incidental hosts, as the virus does not transmit easily between people. The risk of infection increases in endemic areas, where the presence of infected *Culex* mosquitoes, particularly the species *Culex annulirostris*, *Culex annulus*, *Culex fuscocephala*, *Culex gelidus*, *Culex sitiens*, and the *Culex vishnui* species complex, is higher. *Culex tritaeniorhynchus* is the primary vector in most endemic regions. Moreover, JEV has been isolated from over 30 mosquito species across the genera *Aedes*, *Anopheles*, *Armigeres*, *Culex*, and *Mansonia* [79]. Some of these species are also present in Europe and could, in theory, contribute to the transmission of JEV if the virus is introduced [80]. Infection in ardeid birds and pigs is asymptomatic and leads to seroconversion. Ardeid birds migrate seasonally across all continents (except Antarctica), particularly in southern, eastern, central, and northern Asia. These migratory movements may extend JEV’s range [81]. Climate change has been shown to influence the migratory routes of birds and animals and with them the vectors that transmit VBD as already demonstrated in the case of migratory routes of seasonal birds in Korea and the rising cases of Japanese encephalitis virus (JEV) recorded in metropolitan areas [82].

JEV is reported to have originated from an ancestral virus in the Indonesia–Malaysia region. The virus diverged into genotypes I-IV and then spread to over 20 countries across South and Southeast Asia, encompassing an area delimited by Pakistan to the west, the Philippines, Japan to the east, and Australia (Torres Strait islands).

In the last 50 years, the epidemiology of JEV has changed, forming two epidemiological patterns: in temperate regions (northern Thailand and Vietnam, Korea, Japan, China, Nepal, and northern India), the infection occurs during the rainy season and summer; in tropical regions (Southern Thailand and Vietnam, southern India, Sri Lanka, Indonesia, and Malaysia), it is endemic throughout the year at low frequencies. The World Health Organization (WHO) estimates 67,900–100,000 JE cases, with 13,600–20,400 deaths annually. Around 75% of these cases occur in children under 15 years of age [81]. Autochthonous cases of Japanese encephalitis (JE) have never been reported in Europe. However, Platonov et al., in August 2012, reported the detection of an RNA sequence of the JEV NS5 gene in tissue samples from six birds, collected from Tuscany in 2000 and 2003 [83]. The JEV nucleotide sequence found in an Italian pool of *Culex pipiens* mosquitoes was very similar to the JEV RNA sequence found in Italian birds [84]. Probably, unidentified flaviviruses highly similar to JEV or a limited JEV circulation have occurred between birds and mosquitoes in Italy. However, no human cases were reported in Italy. In 2022, it spread to new regions in Australia, and even regions previously considered free of the virus, such as Angola, reported an autochthonous case [85,86].

Currently, there is no specific antiviral treatment for the Japanese encephalitis virus [87].

Vaccination remains the most effective preventive measure against JEV, especially for people living in or traveling to endemic regions in Asia and the Western Pacific. There are several types of JEV vaccines available. However, only three of them are widely used for immunization programs [88]. The WHO recommends vaccination in endemic regions and for travelers spending extended time in these areas. Vaccines have proven to significantly reduce the incidence of JEV, highlighting their critical role in controlling outbreaks and minimizing the disease’s burden on public health [89].

Viruses transmitted by ticks are permanently present in Europe (*Hyalomma marginatum*); the distribution of ticks is shown in Figure 2.

### 3.5. Crimean–Congo Hemorrhagic Fever Virus

Crimean–Congo hemorrhagic fever (CCHF) is a viral disease caused by a tick-borne virus (*Nairovirus*) of the *Bunyaviridae* family, which is endemic in Africa, the Balkans, the Middle East, and Asian countries south of the 50th parallel north—the geographical limit of the principal tick vector. The clinical presentation of Crimean–Congo hemorrhagic fever (CCHFV) is characterized by a range of symptoms that can escalate to severe complications. The course of CCHF can be divided into four phases: incubation, pre-hemorrhagic, hemorrhagic, and convalescent [90].

Crimean–Congo hemorrhagic fever virus (CCHFV) was first identified in human history in 1944 during an outbreak in the Crimea region of the former Soviet Union (initially referred to as Crimean hemorrhagic fever) [91].

The CCHF *orthonairovirus* is typically spread via the bites of ticks of the *Hyalomma* genus. These ticks are widely distributed across Southern and Eastern Europe [92]. *Hyalomma marginatum* has the most prominent role globally in the natural history of CCHF in the Mediterranean basin and Middle Asia [93]. Besides *Hyalomma marginatum*, *Hyalomma aegyptium*, *Rhipicephalus bursa*, and *Rhipicephalus (Boophilus) annulatus* are also involved in transmitting Crimean–Congo hemorrhagic fever (CCHF) in Europe, as indicated by the detection of CCHFV in these tick species in the study conducted in Turkey [94].

In October 2023, *H. marginatum* was reported in Spain, Southern France, Italy, Switzerland, the Balkan region, countries bordering the Black Sea, and African countries along the southern Mediterranean coast. Additionally, the species has been introduced in Germany, Belgium, Austria, the United Kingdom, Norway, Sweden, and Finland [92].

Another CCHFV transmission route is direct contact with patients’ blood and tissues during the acute phase, or with blood or tissues of viremic animals. High-risk exposure exists for people with outdoor activities (for example, soldiers, forest workers, and hikers) and those in close contact with livestock (for example, shepherds, farmers, and veterinarians) [95].

The virus has caused major outbreaks in the EU/EEA neighboring regions, principally in the Balkan region, Turkey, and Russia [96]. One of the biggest European outbreaks was reported in Turkey in 2002 with more than 2500 cases [97].

In the period between 2013 and 2024, 63 cases of CCHFV were reported in Europe (an average of 5.7 cases per year), including autochthonous cases in Bulgaria (51, including five deaths) and Spain (6, including six deaths) and imported cases in the UK (1) and Greece (1), both traveling from Bulgaria.

The majority of cases are due to exposure to the vector during hunting, mountain trips, or farming activities (11 cases). One case was reported as nosocomial (health care provider treating an index case). In the remaining cases, the source of exposure was not known [98]. Several serological surveys of CCHFV were conducted across Europe, revealing the presence of antibodies on livestock in Italy, France, Hungary, and Portugal [99,100,101,102]. Moreover, migratory birds also play a pivotal role in the diffusion of CCHFV-infected ticks, contributing to a potential global diffusion, as demonstrated by Mancuso et al. who detected the CCHFV in a tick collected from a migratory bird in Italy [103]. 

General supportive care with treatment of symptoms is the main approach to managing CCHF in people [104].

Currently, there is no approved vaccine for CCHFV. Several vaccine approaches have been explored, including inactivated, subunit, viral vector-based, mRNA, and DNA vaccines [105].

Prevention is mainly based on reducing contact with ticks and the use of protective clothing and repellents, as well as biosecurity practices in endemic areas and among healthcare workers exposed to infected patients. Effective prevention strategies for Crimean–Congo hemorrhagic fever include improved immunological testing for animals and quarantine, precise surveillance protocols, and early identification and isolation alongside efficient public health policies [106,107,108].

## 4. Pathologies Without Autochthonous Transmission in Humans or Animals but with Presence of Competent Vectors in Europe

The risk of disease introduction in Europe is heightened by the presence of competent mosquito vectors, even in the absence of autochthonous transmission in humans or animals. *Aedes aegypti*, a primary vector for several arboviruses, has demonstrated the capacity to establish in parts of Europe under favorable environmental conditions. While no local transmission has been reported for certain pathologies, the presence of this mosquito species creates a potential for outbreaks if introduced viruses find suitable hosts and ecological conditions. Figure 3 illustrates the distribution of *Aedes aegypti* in Europe, emphasizing regions of particular concern. Monitoring vector populations and assessing their interaction with imported pathogens are critical steps in mitigating the risk of disease emergence.

### 4.1. O’nyong-Nyong Virus

The O’nyong-nyong virus (ONNV) is a mosquito-transmitted *Togaviridae* alphavirus closely related to other members of the Semliki Forest antigenic complex like Semliki Forest virus (SFV), Mayaro virus (MAYV), and CHIKV [109]. Following infection by an ONNV-infected mosquito, symptoms typically appear after an incubation period of around eight days including fever, rash, muscle pain, and debilitating joint pain that can lead to polyarthralgia, although without joint edema—distinct from other alphaviruses like CHIKV and MAYV [109].

ONNV is one of very few arboviruses transmitted by night-feeding anopheline mosquitoes [110]. In particular, *Anopheles gambiae*, *Anopheles funestus*, *and Anopheles stephensi* serve as primary vectors for ONNV but mainly for the *Plasmodium* parasites, leading to overlapping distributions of O’nyong-nyong fever and malaria [111]. Laboratory studies show that *Anopheles gambiae* retain ONNV for 13 days and *Anopheles funestus* retains it for even longer periods (about 20 days) [80,109]. Co-infections with Plasmodium and alphaviruses like ONNV or CHIKV can reduce disease severity in hosts, potentially obscuring the true epidemiological burden of both infections [111,112]. Even though anopheline vectors are widely distributed in Europe, there are no known anopheline mosquitoes for ONNV and, to date, *Anopheles* spp. is the only recognized vector for ONNV [113]. In experimental studies, it was demonstrated that the culicine mosquito species *Aedes aegypti*, recently introduced in some parts of Europe, might be a competent vector for ONNV [114,115].

It was first isolated in 1959 during a widespread East and Central African outbreak affecting over 2 million people [116]. Following its initial epidemic, ONNV re-emerged in southern Uganda in 1996 after a prolonged absence [117]. As of this review, ONNV cases remain geographically limited to Africa, apart from sporadic imported cases reported in Europe [118].

Currently, there is no specific treatment for ONNV. No commercially available vaccine exists for ONNV, though the recent FDA-approved live attenuated CHIKV vaccine induces strong seroprotective responses and may offer cross-protection against related viruses, including ONNV [109]. Antibody-based therapies remain in experimental stages, with studies highlighting the cross-neutralizing potential of monoclonal antibodies (mAbs) like CHK-265, which is effective against several alphaviruses, including ONNV, in mouse and non-human primate models [119,120]. For antiviral approaches, emerging strategies focus on non-toxic compounds that interrupt virus–host interactions, such as FHL1 and G3BP1 inhibitors, offering promising avenues for drug development [121].

### 4.2. Rift Valley Virus

Rift Valley fever (RVF) is an acute viral zoonosis primarily affecting domestic and wild ruminants, camels, and humans. It is caused by the Rift Valley fever virus (RVFV) belonging to the *Phlebovirus* genus (Bunyaviridae family) [122].

Even though humans are often asymptomatic, severe manifestations can occur during large outbreaks in which thousands of people may become infected. Acute hepatitis, retinitis, encephalitis, and other neurological signs, as well as hemorrhagic diathesis, may be present [123].

Several routes of transmission have been identified: via vectors such as mosquitoes, especially those of the genera *Aedes* and *Culex*, but also *Anopheles*, *Coquillettidia*, *Culiseta*, *Eretmapodites*, and *Mansoni* [124,125], as well as other arthropod vectors, including ticks and flies [123]. Transmission can also occur through direct contact with the blood, body fluids, or tissues and organs of infected animals and aborted animal fetuses or infected fomites, as well as vertical transmission both in humans and animals [124]. Noteworthy, RFVF is also maintained in *Aedes* spp. populations through trans-ovarial transmission. Inhaling aerosols of infectious body fluids and consuming raw or unpasteurized milk have also been identified as risk factors for RVFV infection [125].

Considering all the routes of transmission and its contagiousness, RVF has the potential to spread easily to other countries reaching epidemic levels, and for these reasons, it is defined as a transboundary animal disease (TAD) [126]. TADs pose a significant threat due to their economic, trade, and food security impact across numerous countries, resulting in negative One Health outcomes. Their control and management, often requiring exclusion measures, necessitate cooperation among multiple nations, a high surveillance system together with response capacities [127].

Since its first identification in 1931 in a sheep on a farm in the Rift Valley of Kenya, the disease has spread across the African continent and into the Arabian Peninsula (e.g., Saudi Arabia and Yemen) [128,129]. Here, it has become endemic, with periodic epidemics occurring at intervals of 5–15 years, resulting in the death of thousands of livestock animals, during which most human infections occur [130]. In particular, until 1975, RVF was regarded as an African animal disease producing high mortality rates in newborn ruminants, especially sheep and goats, and abortion in pregnant animals, while human cases were rare and with mild clinical manifestations. From that moment on, severe outbreaks in humans were reported in the African continent (South Africa 1975, Egypt 1977, Mauritania 1987, and eastern Africa 1997), and outside Africa (Saudi Arabia and Yemen 2000), becoming a serious concern [123].

To date, no outbreaks have been reported in Europe [131], but after the recent outbreaks in Mayotte, a French overseas department in the Indian Ocean (2018–2019) [132] and some seropositive cases, both in human and animal populations detected in Turkey, Tunisia, Egypt, Libya, and Algeria, raised the attention of the EU for a possible incursion from neighboring countries [133,134,135,136]. The transport of infected animals and vectors via flights, shipping containers, or road transport is considered a plausible pathway for introduction into the EU, and surveillance has to be strengthened.

### 4.3. Yellow Fever Virus

*Yellow fever* virus (YFV) is a member of the *Flaviviridae* family transmitted by the bites of infected mosquitoes to humans from non-human primate (NHP) reservoirs [137].

Yellow fever can range from mild to fatal disease and it is characterized by three clinical stages: After an incubation of 3–6 days, the viremic period (first stage) starts with a sudden onset of fever, headache, malaise, myalgia, nausea, and vomiting. During these 3–5 days, the blood is infectious to anthropophilic biting mosquitoes. After a remission period of 1–2 days (second stage), a small percentage of patients (15–25%) progress to the intoxication phase (third stage), where viremia is reduced in favor of an increased humoral-mediated reaction responsible for marked physical illness with signs of liver and renal failure, and spontaneous hemorrhages [138].

Among vectors, *Ae. aegypti* is the highest effective vector for YF, and it is responsible for large epidemics in tropical cities, where is ubiquitous [12]. This culicid is not established in the continental area of the European Union (EU), and therefore, the risk of autochthonous transmission from travel-related cases is low. However, *Ae. albopictus*, largely distributed in the Mediterranean parts of the EU, has been proposed as having the potential to transmit YFV, as demonstrated experimentally [139,140].

Even though epidemiological and genetic studies sustain the hypothesis that the YF virus originated in Africa, it was first discovered in the Americas at the end of the 15th century, where it emerged following the slave trade [141,142]. In natural forest habitats, YFV is transmitted between NHPs through mosquitoes (*Haemogogus* spp. and *Sabethes* spp. in South America; *Aedes* spp. in Africa) in a cycle known as sylvatic transmission [143,144]. Humans can accidentally become infected if they enter forest areas to hunt for and gather food and are bitten by infected mosquitoes, or if infected mosquitoes that previously fed on a viremic monkey (jungle yellow fever) move into human-populated areas seeking blood [145]. Once infected humans in the forest bring the virus to urban areas, highly anthropophilic mosquitoes (*Aedes* spp.) can spread it quickly, shifting the transmission from the sylvatic to the urban cycle (urban yellow fever). In densely populated tropical regions, urban transmission can lead to significant epidemics and even pandemics, though sometimes transmission is low enough just to maintain the virus in the population [146]. A mixed cycle, the so-called peri-urban cycle, can be sustained when sylvatic and domestic vector species and humans live or work in jungle border areas, and it is mostly observed in Africa (usually in the savannah) [147].

No autochthonous cases were ever recorded in the EU, but several travel-associated YF cases have been reported among unvaccinated European travelers to endemic areas, in particular, in 7 cases from 1999 to 2017 and 13, including 1 fatal case, reported in 2018 [148,149].

No approved antiviral drugs against YFV are currently available, although both in vitro and in vivo studies demonstrated that sofosbuvir, a direct antiviral agent used for hepatitis C virus, might be used to treat YFV infection, due to its capacity to reduce the viral load and improve biochemical markers [150,151,152].

An effective vaccine based on the live attenuated YFV-17D virus remains the most useful weapon to prevent YFV infection, because it confers long-lasting protection against the disease, in both immunocompromised and healthy individuals [153]. Clinicians and health care personnel should promote the YFV vaccine among international travelers, in order to prevent severe illness and to avoid the onset of an outbreak in non-endemic but high-risk places [154]. Even though the risk of autochthonous transmission of YFV in the EU from travel-imported cases is negligible, the surveillance and awareness have to be high due to climate changes, the increased flow of migration from endemic areas and/or travelers, and the potential establishment of effective vectors [155].

## 5. Pathologies Without Autochthonous Transmission in Humans or Animals and Without the Presence of Competent Vectors in Europeviruses Transmitted by Midges (*Culicoides* spp.)

Awareness of pathologies without autochthonous transmission in humans or animals and without the presence of competent vectors in Europe is essential for proactive public health measures. While diseases like the Oropouche virus currently lack established transmission in the region, global travel, trade, and climate change could facilitate the introduction of both pathogens and vectors. Understanding these risks allows for early detection, targeted surveillance, and the development of contingency plans to prevent potential outbreaks and safeguard public health.

### Oropouche Virus

The emerging arbovirus Oropouche virus (OROV) has recently gained attention, though awareness and understanding of its impact remain limited. OROV belongs to the *Bunyavirales* order, *Peribunyaviridae* family, and Orthobunyavirus genus, characterized by a negative-sense, single-stranded RNA and a spherical lipid envelope [156,157]. Within the Orthobunyavirus genus, OROV is part of the Simbu serogroup, which includes 25 viruses across two phylogenetic subclades: Manzanilla–Oropouche (subclade A) and Simbu-related viruses (subclade B) [158]

Following the bite of an OROV-infected midge or mosquito, a 3–8-day incubation period typically precedes the onset of symptomatic illness [159]. Infected individuals experience an acute febrile illness with symptoms such as fever, headache, myalgia, and joint pain, and about 60% may see a recurrence of symptoms within one to two weeks While most cases last 2–7 days, some develop more severe manifestations, including hemorrhagic or neurological symptoms, particularly in cases involving central nervous system (CNS) infection [160].

OROV is primarily transmitted to humans through bites from infected *Culicoides paraensis* midges, typically found in forested and aquatic areas [161]. However, while *Culicoides* midges are considered the primary vectors for OROV, other insects such as *Culex quinquefasciatus* and *Aedes serratus* mosquitoes may also serve as potential vectors [162]. Vector competence for OROV, however, remains insufficiently studied, particularly regarding the virus’s ability to overcome the midgut barrier in these vectors [163]. OROV was first identified in 1955 during a febrile illness outbreak in Trinidad and Tobago and has since caused epidemics in several South and Central American countries, including Brazil, Peru, and Panama [164,165,166]. Over 500,000 cases have been diagnosed to date, though the actual number is likely higher due to underreporting and clinical similarity to other co-circulating arboviruses [160]. In May 2024, Cuba reported its first outbreak, marking a notable spread beyond the typical geographic range [167]. As of writing this article, no autochthonous cases or vector presence have been documented in Europe, but only 19 imported cases were recorded in Spain (12), Italy (5), and Germany (2), mainly comprising travelers returning from Cuba and Brazil [168]. Furthermore, recent cases in European travelers returning from Cuba underscore OROV’s potential for global spread amid increased human mobility and environmental change [169,170].

Currently, there is no specific treatment or vaccine available for OROV, and management relies on symptom control and supportive care [159]. Several vaccine candidates, including inactivated virus, live attenuated virus, and recombinant protein-based vaccines, have shown promise in preclinical trials, with some progressing to clinical testing stages [157]. Research into OROV vaccine development has also explored cross-protection strategies informed by vaccine designs for related viruses like Schmallenberg and Akabane, with promising results in veterinary and animal models that may provide a foundation for OROV vaccine strategies [171].

## 6. Conclusions

Vector-borne disease epidemiology is influenced by complex interactions between climate, wildlife, domestic animals, humans, and arthropod vectors. Over recent decades, the spread of vectors of arboviruses within Europe has been modified by factors such as climate change and land-use alterations [172,173], and the study of vector ecology has become critical for understanding VBD patterns due to the broader geographic distribution of arboviruses and more frequent outbreaks among humans and animals [174]. Additionally, the increase in global travel, including tourism travel and trade as well, has exposed Europe to a higher risk of introducing exotic arthropod vectors and local transmission of arboviruses like DENV, CHIKV, ZIKV, and CCHFV from tropical and subtropical regions [175].

The establishment of integrated arbovirus surveillance programs is an essential strategy to mitigate the risks associated with the emergence and spread of these pathogens [176,177]. Such programs should prioritize the implementation of comprehensive control measures, including wildlife monitoring and domestic animal surveillance, to detect and address potential reservoirs and transmission routes. Reducing the interactions between wildlife and domestic animals, alongside the adoption of rigorous disinfection protocols, can significantly minimize the risk of spillover events. To further safeguard human health, it is critical to avoid human transmission through blood, tissue, and organ donations by screening all these biological samples for the presence of arbovirus pathogens. However, detecting emerging outbreaks is challenging and demands a high level of vigilance and laboratory capability, particularly for the less recognized but potentially dangerous pathogens.

Vector control is another critical component in managing arbovirus transmission. Strategies such as reducing breeding sites, deploying insecticides responsibly, and promoting the use of personal protective measures like insect repellent can substantially lower vector populations and limit human exposure. Incorporating community-based initiatives to raise awareness about vector habitats and control practices further enhances the effectiveness of these measures.

Equally important is the development and deployment of effective vaccination programs, which can offer a proactive defense against arboviral infections, especially in high-risk regions. Public awareness campaigns and community engagement should accompany these measures to enhance preventive practices and reduce vector exposure. Strengthening diagnostic infrastructure and fostering international collaboration are pivotal to identifying less-recognized but potentially dangerous pathogens.

By integrating surveillance, vaccination, and preventive strategies, we can build a resilient framework to protect One Health from the threats posed by climate change.

While it is clear that climate change will impact VBD risk, it is also recognized that it is not the only factor that plays an important role. Other equally important elements include socioeconomic development, urbanization, land-use change, climate change adaptation, migration, and globalization. As a result, assessing the public health risks posed by climate change is complex.

However, by adopting a One Health approach, Europe can be better equipped to handle the challenges posed by vector-borne diseases in a changing climate and it can decrease the health defects of these diseases. This comprehensive strategy not only aims to safeguard human health but also considers the well-being of animals and the environment, leading to more sustainable and effective solutions. Only an interdisciplinary approach can enable the development of comprehensive veterinary and public health interventions, community support systems, and policy measures necessary to deal with these (re-)emerging infectious diseases.

Collaborative research efforts that actively integrate knowledge from medical and veterinary sciences, climate sciences, and social sciences are needed to develop a comprehensive understanding of the multifaceted impacts of climate change in different geographical contexts.

In conclusion, our review may have some important implications for the community, particularly in addressing the increasing risk of vector-borne diseases (VBDs) in Europe. Integrated surveillance programs and proactive control measures might benefit communities by preventing outbreaks and ensuring that health authorities are better prepared to respond quickly to emerging threats

## Figures and Tables

**Figure 1 pathogens-14-00063-f001:**
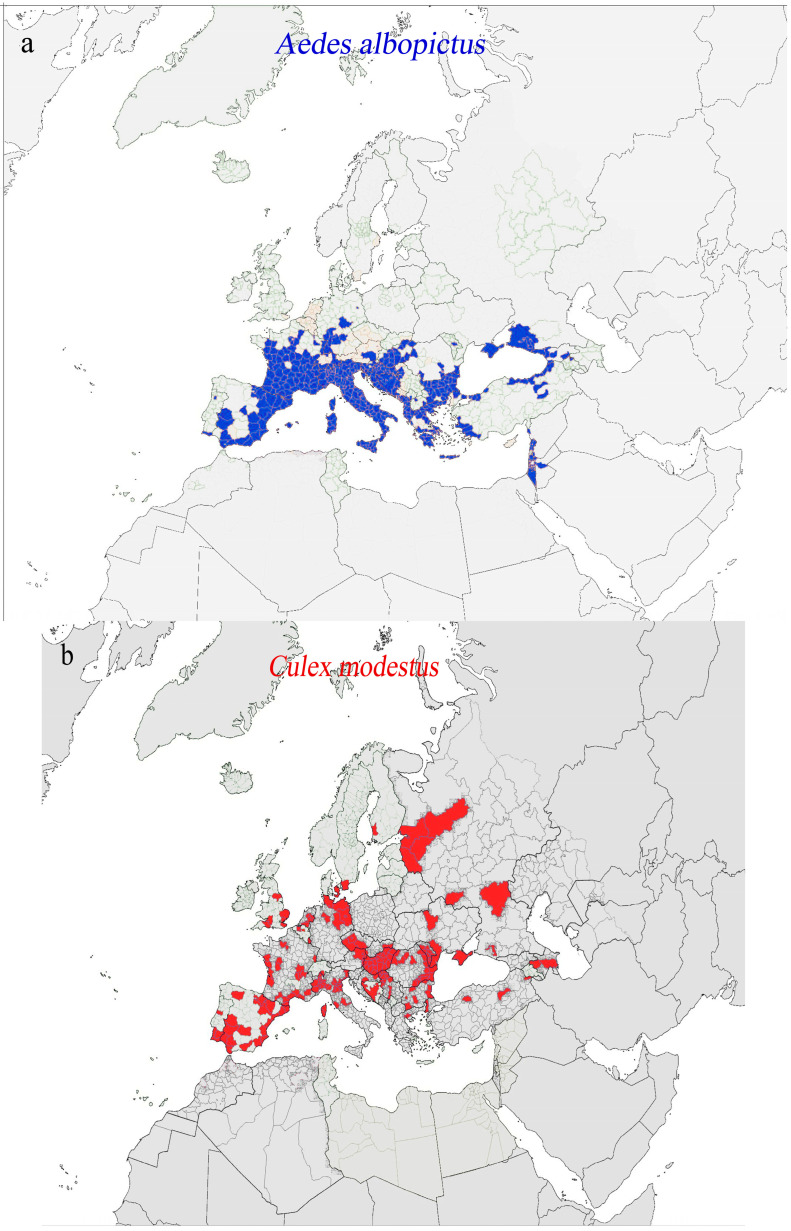
Mosquito distribution in Europe according to ECDC source [9]. (**a**) *Aedes albopictus*; (**b**) *Culex modestus*; (**c**) *Culex theileri*; (**d**) *Culex perexiguus unvittatus*; (**e**) *Culex pipiens*.

**Figure 2 pathogens-14-00063-f002:**
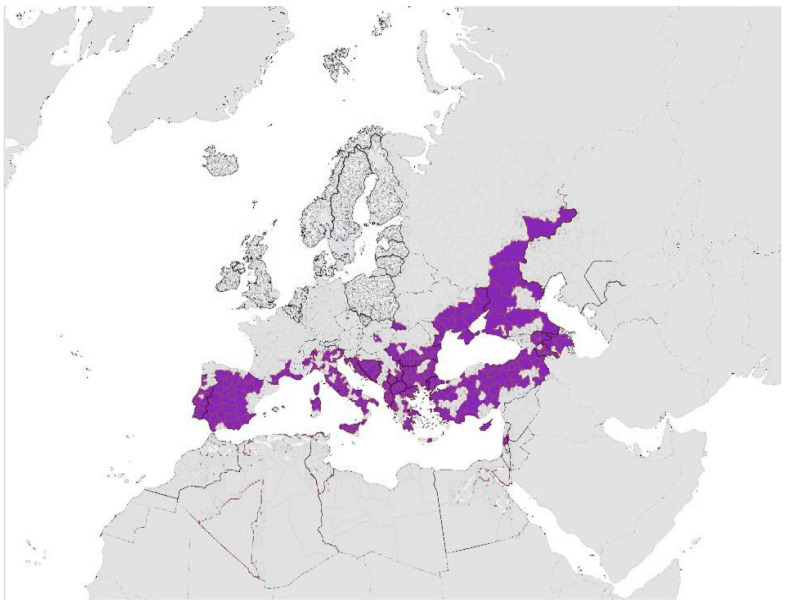
Distribution of *Hyalomma marginatum* (purple color) in Europe according to ECDC source (available online: Hyalomma marginatum—current known distribution: October 2023; accessed on 20 November 2023).

**Figure 3 pathogens-14-00063-f003:**
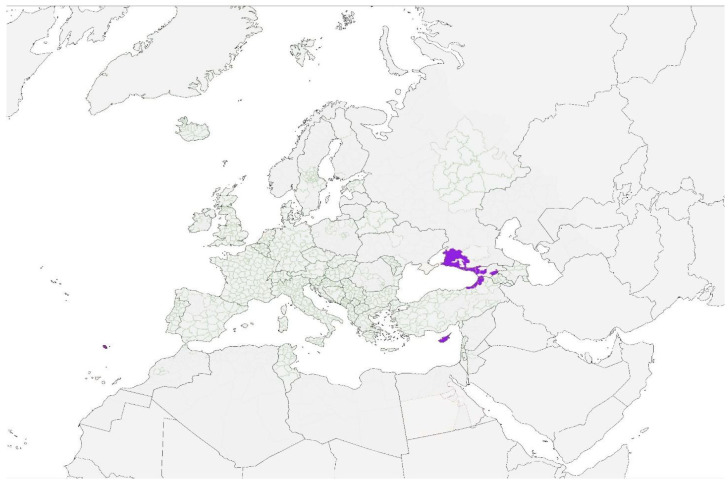
Distribution of *Aedes aegypti* (purple color) in Europe according to ECDC source [9].

**Table 1 pathogens-14-00063-t001:** A summary of the autochthonous and imported cases of VBDs in both humans and animals across Europe along with the corresponding vectors and their distribution.

Pathologies with Active Transmission and Autochthonous Cases in Human or in Animals Transmitted by Vectors Permanently Present Across Europe				
DISEASE	VIRUS	CASES IN EUROPE	VECTORS	VECTOR DIFFUSION
Chikungunya fever	Chikungunya virus	In human: Southern France (2010, 2014), Italy (2007, 2017)	*Ae. albopictus* *Ae. aegypti*	*Ae. a lbopictus*: widely diffused *Ae. a egypti*: Cyprus, Eastern Black Sea, Madeira
Dengue fever	Dengue virus	In human: since 2010, constant circulation in the Mediterranean basin	*Ae. albopictus* *Ae. aegypti*	*Ae. a lbopictus*: widely diffused *Ae. a egypti*: Cyprus, Eastern Black Sea, Madeira
Zika fever	Zika virus	In human: Southern France (2019)	*Ae. albopictus* *Ae. aegypti*	*Ae. a lbopictus*: widely diffused *Ae. a egypti*: Cyprus, Eastern Black Sea, Madeira
Japanese encephalitis	*Japanese encephalitis* virus	In animal: birds in Italy (2012)	*Culex tritaeniorhynchus* (also *Culex annulirostris*, *Culex annulus*, *Culex fuscocephala*, *Culex gelidus*, *Culex sitiens*, and *Culex vishnui*)	*Culex tritaeniorhynchus*: Albania, Greece, Turkey, and the Middle East Mediterranean
Crimean–Congo hemorrhagic fever	Crimean–Congo hemorrhagic fever virus	In human: Turkey, Spain, Bulgaria, Russia In animals: seropositive cases in bovines in Italy	*Hyalomma marginatum* (also *Hyalomma aegyptium*, *Rhipicephalus bursa,* and *Rhipicephalus* (*Boophilus*) *annulatus*))	*Hyalomma marginatum*: Spain, Southern France, Italy, Switzerland, the Balkan region, countries bordering the Black Sea, Russia
**Pathologies Without Autochthonous Transmission in Humans or Animals but with Presence of Competent Vectors in Europe**				
**DISEASE**	**VIRUS**	**CASES IN EUROPE**	**VECTORS**	**VECTOR DIFFUSION**
O’nyong-nyong fever	O’nyong-nyong virus	Sporadic imported cases reported in Europe	*Anopheles gambiae*, *A. funestus*, and *A. stephensi*.	Only *Anopheles maculipennis* is widely diffused (theoretically competent).
Rift Valley fever	Rift Valley virus	In human: only seropositive cases in Turkey In animals: seropositive cases in Turkey	*Aedes* spp., *Culex* spp. (also *Anopheles*, *Coquillettidia*, *Culiseta*, *Eretmapodites*, and *Mansoni*)	Widely diffused
Yellow fever	Yellow fever virus	In human: 20 travel-associated cases (1999–2018)	*Ae. Albopictus* and *Ae. aegypti*	*Ae. a lbopictus*: widely diffused *Ae. a egypti*: Cyprus, Eastern Black Sea, Madeira
**Pathologies Without Autochthonous Transmission in Humans or Animals and Without Presence of Competent Vectors in Europe**				
**DISEASE**	**VIRUS**	**CASES IN EUROPE**	**VECTORS**	**VECTOR DIFFUSION**
Oropouche fever	Oropouche virus	In human: 19 recent travel-associated cases	*Culicoides paraensis*	No diffusion in Europe

## Data Availability

Not applicable.

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
