# Peer review of "Introduction of Vector-Borne Infections in Europe: Emerging and Re-Emerging Viral Pathogens with Potential Impact on One Health"

_pathogens, 2025, doi:10.3390/pathogens14010063_

Round 1
Reviewer 1 Report
Comments and Suggestions for Authors
Manuscript review
Research Article: Pathogens-3272616 - Introduction of Vector-Borne Infections in Europe: Emerging and Re-Emerging Viral Pathogens with Potential Impact on One Health
Summary of data presented
The article consists of a review that sheds light on the introduction and re-emergence of vector-borne infections in Europe, driven by factors such as climate change, globalization, and ecological disruptions, which present a growing public health concern. This review focuses on arboviruses transmitted by mosquitoes (e.g., Dengue, Zika, Chikungunya) and ticks (e.g., Crimean-Congo hemorrhagic fever), examining their classification based on local transmission status, vector presence, and reported cases. The expansion of vectors like Aedes albopictus and increased human mobility have heightened the risk of disease spread, underscoring the necessity of integrated One Health strategies. Current responses include enhanced surveillance, public awareness campaigns, and outbreak management, but challenges such as limited antiviral therapies and asymptomatic cases hinder containment. The study highlights the need for combined climate modelling, disease monitoring, and public health interventions to address Europe's evolving threat of vector-borne diseases.
I appreciate the opportunity to review this article and commend the authors for addressing the crucial and timely topic of managing vector-borne diseases through a One Health approach. The subject matter holds significant merit and relevance, especially given its implications for public health and environmental policy. However, I believe that revising certain aspects of the text could enhance the clarity and flow of the presentation, ensuring that the valuable insights offered are conveyed in a more fluid and organized manner.
Please consider revising the following points:
Main points to review
General comments:
The article is long, and as a result, reading it becomes slow and difficult. I believe that the most significant problems faced by authors at the time of writing are organizing ideas concisely and, above all, delimiting or approaching the primary objective expressed in the title more directly. Furthermore, the article was submitted with severe formatting and fundamental punctuation flaws.
Abstract:
1. The link between the ideas can be improved. Additionally, the definition of TBDs is missing, impacting the audience's comprehension.
2. Check for double spaces between words.
3. Lines 17-18: This statement could be improved. The example of arboviruses should come first than transmitted by mosquitoes.
4. Lines 20-23: This statement can be rewritten.
5. Lines 23-25: For me, the idea of One Health can be explored in the beginning, and the date of the first cases (or decade) could be included to give an overview.
Introduction:
General comment: Overall, I enjoyed reading the introduction. The authors included all the information necessary to provide an expensive background about the arboviruses' situation, as well as the features associated with its increase and spreading. However, I suggest reviewing the link between the statements so it can be refined.
1. Lines 39-41: I agree with the idea of using COVID-19 to explain the importance of the emergence of new pathogens and global spreading. However, we have a lot of examples regarding the alarming scenario of arboviruses worldwide, as well as the increase in severity and number of cases. Perhaps the authors should use these ideas to interconnect with the climate change topic. Maybe even start the article by bringing the climate change first and then using the arboviruses scenario.
2. Lines 52-53: I suggest using this statement to make a connection with the following paragraph instead of leaving a single phrase alone.
3. Table 1: Only the species should be in italics. I suggest re-checking the table.
Session 2:
General Comment: I suggest the authors review and clarify the criteria and the type of documents/articles accessed. Additionally, the language can be refined.
1. Lines 70: The section's title doesn't need to be written in uppercase.
2. Lines 71-75: Check formatting.
3. Lines 76-82: The two paragraphs can be joined, and the ideas can be better expressed.
4. Lines 83-84: It is unclear why the authors divided into the categories presented. Furthermore, the categories represented by viruses may or may not already be covered/associated/included in the other disease categories?
Section 3:
General Comment:
1. Lines 121-122: I suggest writing a paragraph introducing the topic. As well as it's missing a final point. The same for section 4. Pathologies without autochthonous transmission in humans or animals but with the presence of competent vectors in Europe (Lines 580-581) and 5. Pathologies without autochthonous transmission in humans or animals and without the presence of competent vectors in Europe Viruses transmitted by midges (Culicoides spp) (lines 733-735).
2. Figure 1: I suggest the authors do a single figure (and bigger) and select different colours to represent the presence or absence of the vectors. Maybe divide into categories (single or multiple vectors). Using the same colour makes it difficult to differentiate which one it is.
Specific viruses sections:
General Comment:
1. The paragraphs can be better linked.
2. Lack of formatting and several double spaces between words.
3. To contextualise the virus, The authors could condense all information (origin, strains, symptoms, diagnosis) into one or two paragraphs maximum.
4. The topic as a whole lacks a logical chronology; the authors include data about the viral structure and classification (lines 174-180) that should be presented earlier.
5. The article is a review of the introduction and spread of these pathogens in European countries; however, the story could be told in a more concise and direct way, providing only essential data about the introduction and interconnection with countries outside the European continent.
6. Reading becomes impaired and tiring due to the lack of sequential organization of ideas.
7. I suggest including, if the authors wish, phylogeographic articles in the literature review methodology, where they seek to describe the origin and dissemination routes of arboviruses in Europe.
8. During the writing process, we usually do not leave just one sentence as a paragraph. These sentences should be included within more extensive paragraphs.
Conclusion:
General Comment: Lack of references to support the authors ideas. If there is no article, I suggest searching for articles on the same topic applied to other countries or regions across the globe. Additionally, would be interesting if the authors could include a paragraph about the relevance of their work or how the community can be beneficiated by the information reported in the paper.
1. Lines 781: VBDs should be entirely written in the beginning of the paragraph.
2. Lines 786-789: Include the reference.
3. Lines 790-795: Authors should include how it's done or if there are measures to avoid transmission of these viruses (blood/organ donors serological screening, for example).
Author Response
General comments:
The article is long, and as a result, reading it becomes slow and difficult. I believe that the most significant problems faced by authors at the time of writing are organizing ideas concisely and, above all, delimiting or approaching the primary objective expressed in the title more directly. Furthermore, the article was submitted with severe formatting and fundamental punctuation flaws.
Abstract:
- The link between the ideas can be improved. Additionally, the definition of TBDs is missing, impacting the audience's comprehension.
Thank you for the suggestion. We have reorganized the Abstract and the abbreviations TBD has been deleted
- Check for double spaces between words.
We have corrected the formatting errors
- Lines 17-18: This statement could be improved. The example of arboviruses should come first than transmitted by mosquitoes.
We have performed the modification
- Lines 20-23: This statement can be rewritten.
We have deleted this statement
- Lines 23-25: For me, the idea of One Health can be explored in the beginning, and the date of the first cases (or decade) could be included to give an overview.
We have performed the modifications as suggested by the Reviewer
Introduction:
General comment: Overall, I enjoyed reading the introduction. The authors included all the information necessary to provide an expensive background about the arboviruses' situation, as well as the features associated with its increase and spreading. However, I suggest reviewing the link between the statements so it can be refined.
- Lines 39-41: I agree with the idea of using COVID-19 to explain the importance of the emergence of new pathogens and global spreading. However, we have a lot of examples regarding the alarming scenario of arboviruses worldwide, as well as the increase in severity and number of cases. Perhaps the authors should use these ideas to interconnect with the climate change topic. Maybe even start the article by bringing the climate change first and then using the arboviruses scenario.
We appreciate the comment. Thanks for the suggestion. In the introduction, we have placed more emphasis on climate change and its relationship with the transmission of vector-borne diseases (VBDs), followed by a deeper exploration of the epidemic transmission dynamics of selected VBDs.
- Lines 52-53: I suggest using this statement to make a connection with the following paragraph instead of leaving a single phrase alone.
Thanks for the suggestion. We have revised the entire paragraph to ensure greater consistency with the topics discussed.
- Table 1: Only the species should be in italics. I suggest re-checking the table.
The Table 1 has been corrected
Session 2:
General Comment: I suggest the authors review and clarify the criteria and the type of documents/articles accessed. Additionally, the language can be refined.
- Lines 70: The section's title doesn't need to be written in uppercase.
- 2. Lines 71-75: Check formatting.
- Lines 76-82: The two paragraphs can be joined, and the ideas can be better expressed.
- Lines 83-84: It is unclear why the authors divided into the categories presented. Furthermore, the categories represented by viruses may or may not already be covered/associated/included in the other disease categories?
Thanks for the suggestion. We have clarified the criteria used for the selection of diseases and documents accessed. Moreover, we have changed the uppercase titles, the formatting, we have joined the two paragraphs as proposed and we have specified why we have grouped the arboviral infections in the different categories.
Section 3:
General Comment:
- Lines 121-122: I suggest writing a paragraph introducing the topic. As well as it's missing a final point. The same for section 4. Pathologies without autochthonous transmission in humans or animals but with the presence of competent vectors in Europe (Lines 580-581) and 5. Pathologies without autochthonous transmission in humans or animals and without the presence of competent vectors in Europe Viruses transmitted by midges (Culicoides spp) (lines 733-735).
Thank you for giving us the opportunity to improve the work. As suggested, we have added a specific introduction for each paragraph, highlighting the principal aspect of different groups.
- Figure 1: I suggest the authors do a single figure (and bigger) and select different colours to represent the presence or absence of the vectors. Maybe divide into categories (single or multiple vectors). Using the same colour makes it difficult to differentiate which one it is.
Designing this figure was extremely difficult because inserting all the vectors in a single map was complicated by the overlapping of these vectors in the same geographical area. The same difficulty we would encounter in presenting with different colors geographical areas with absence, presence of single or multiple vectors because we could have areas in which 2 or 3 vectors coexist, and we would not be able to distinguish them. To make it easier to understand, we put the name of the vector inside the figure instead of in the caption and used different colors for the various vectors
Specific viruses sections:
General Comment:
- The paragraphs can be better linked.
According to the reviewer’s suggestion, we have reorganized these sections
- Lack of formatting and several double spaces between words.
We have checked for all these types of errors
- To contextualise the virus, The authors could condense all information (origin, strains, symptoms, diagnosis) into one or two paragraphs maximum.
According to the reviewer’s suggestion, we have reduced these informations
- The topic as a whole lacks a logical chronology; the authors include data about the viral structure and classification (lines 174-180) that should be presented earlier.
We have corrected all these not suitable sequences
- The article is a review of the introduction and spread of these pathogens in European countries; however, the story could be told in a more concise and direct way, providing only essential data about the introduction and interconnection with countries outside the European continent.
We have tried to underline these aspects
- Reading becomes impaired and tiring due to the lack of sequential organization of ideas. I suggest including, if the authors wish, phylogeographic articles in the literature review methodology, where they seek to describe the origin and dissemination routes of arboviruses in Europe.
We have reorganized these sections describing the introduction of vectors and the presence of outbreaks in a sequential order
- During the writing process, we usually do not leave just one sentence as a paragraph. These sentences should be included within more extensive paragraphs.
We have followed this suggestion and we have tried to have only one single sentence.
Conclusion:
General Comment: Lack of references to support the authors ideas. If there is no article, I suggest searching for articles on the same topic applied to other countries or regions across the globe. Additionally, would be interesting if the authors could include a paragraph about the relevance of their work or how the community can be beneficiated by the information reported in the paper.
- Lines 781: VBDs should be entirely written in the beginning of the paragraph.
It has been corrected
- Lines 786-789: Include the reference.
References have been included
- Lines 790-795: Authors should include how it's done or if there are measures to avoid transmission of these viruses (blood/organ donors serological screening, for example)
We have included briefly these measures
Reviewer 2 Report
Comments and Suggestions for Authors
Reviewer report
ID: pathogens-3375623
Title: Introduction of vector borne infections in Europe: emerging and re-emerging viral pathogens with potential impact on One Health
General comments:
The review is good and gives basic information about some viruses threating human health.
Specific comments:
The Bhanja virus, West Nile virus, Usutu virus and Tick borne encephalitis virus infections needs to be included in both human and animals in Table 1 and in the text.
The word “imported” was written mistakenly as “mported” in Table 1 heading.
Maps needs to be organised for each diseases investigated so far as the authors did for mosquitos. These maps can be added as supplementary materials.
References should be taken from the actual authors who performed the work rather than taking from review papers.
There are english errors needs to be corrected like “reference”-“References”
A new heading “Determinants of the VBDs” needs to be written in few pages as well as zoonotic relationship. Role of animals.
A new heading for organisations for VBDs in terms of whether these organisations are doing what ?
A seperate topic for Control strategies

Author Response
General comments:
The review is good and gives basic information about some viruses threating human health.
Specific comments:
The Bhanja virus, West Nile virus, Usutu virus and Tick borne encephalitis virus infections needs to be included in both human and animals in Table 1 and in the text.
We thank Reviewer 2 for the opportunity to clarify this aspect. We acknowledge that the list of emerging pathogens in Europe is extensive, and our review includes a non-exhaustive selection of vector-borne and tick-borne viral diseases. Given the complexity of the topic, we chose to focus on viral infections listed by the WHO in their updated report as of September 26, 2024, accessible via the following link: https://www.who.int/news-room/fact-sheets/detail/vector-borne-diseases.
For this reason, Bhanja virus and Usutu virus were not included in our review. Furthermore, Usutu virus, alongside West Nile virus and tick-borne encephalitis, is already considered endemic in Europe. We have detailed this point in the selection criteria section (lines 94–95).
The word “imported” was written mistakenly as “mported” in Table 1 heading.
The mistake has been corrected
Maps needs to be organised for each diseases investigated so far as the authors did for mosquitos. These maps can be added as supplementary materials.
Thank you for the suggestion, but we believe that such maps perhaps can overlap with the data presented in the Table 1 where the list of the countries in which disease outbreaks occurred have already been described in detail
References should be taken from the actual authors who performed the work rather than taking from review papers.
Thank you, we have corrected all the references.
There are english errors needs to be corrected like “reference”-“References”
Thank you, we have corrected the mistake.
A new heading “Determinants of the VBDs” needs to be written in few pages as well as zoonotic relationship. Role of animals.
Thank you for the comment and suggestion. We have added a few lines regarding the determinants of vector-borne diseases (VBDs). Similarly, we have included additional information about the role of animals in VBD transmission, providing examples of zoonotic transmission and discussing the anthropological impact on these processes. We did not consider it functional to delve further beyond the revised paragraph, in order to avoid overburdening the readability of the article. (from 112)
A new heading for organisations for VBDs in terms of whether these organisations are doing what ?
Thank you for this comment. We interpreted this suggestion as an opportunity to mention some health surveillance programs and the integrated management of recent outbreaks
A separate topic for Control strategies
We have included control strategies in the conclusion chapter, in order to not lengthen the text. We have mostly focused on integrated surveillance, vector controls and vaccine strategies both for human and animal population.
Reviewer 3 Report
Comments and Suggestions for Authors
The topic understudy is the emergence of vector borne diseases in Europe and a One Health approach to control. However, the authors spent a great deal of time on the clinical presentation of the diseases under consideration which detract from the topic at hand. It would be more helpful if the focus on epemiology, control (including but not vaccination).
The paper does not mention the role of increased travel associated with tourism to endemic area on the increased risk to introducing VBD into Europe and failed to review travel hotspots such at the Caribbean. In fact, there is no review of CHKV in the Caribbean where there have been major outbreaks.
Comments on the Quality of English LanguageThe paper should be revised to improve the quality of English.
Author Response
Comments and Suggestions for Authors
The topic understudy is the emergence of vector borne diseases in Europe and a One Health approach to control. However, the authors spent a great deal of time on the clinical presentation of the diseases under consideration which detract from the topic at hand. It would be more helpful if the focus on epemiology, control (including but not vaccination).
Thank you for the suggestion. We have reduced the information about the clinical point of the virus diseases and have introduced a section about control strategies
The paper does not mention the role of increased travel associated with tourism to endemic area on the increased risk to introducing VBD into Europe and failed to review travel hotspots such at the Caribbean. In fact, there is no review of CHKV in the Caribbean where there have been major outbreaks.
We have introduced few sentences about the role played by the tourism as possible risk to introduce VBDs in Europe in the Conclusion section and inserted the outbreak in Caribbean islands in CHIKV section
Comments on the Quality of English Language
The paper should be revised to improve the quality of English.
We have revised the language
Reviewer 4 Report
Comments and Suggestions for Authors
MS-NO- pathogens-3375623
Title: Introduction of Vector Borne Infections in Europe: Emerging and Re-Emerging Viral Pathogens with Potential Impact on One Health
Abstract
This section is fine.
Introduction
Page 2 paragraph 3: Please insert appropriate reference “In Europe, climate change has created favorable microenvironments for mosquitoes and ticks in areas previously considered at low-risk”
Table 1.
Please correct the sentence “Pathologies without autochthonous transmission in humans or animals and without the presence of competent vectors in Europe” as “Pathogens without autochthonous transmission in humans or animals and without the presence of competent vectors in Europe”
It is good if authors provide information how climate change, globalization, wild life and ecological disruptions increase the introduction, distribution and transmission of arthropod-borne arbovirus in Europe.
CONCLUSIONS
This section seems good
References
Authors should avoid citing articles published in predatory journals and replace them with references from high-quality international journals.
Author Response
Abstract
This section is fine.
Introduction
Page 2 paragraph 3: Please insert appropriate reference “In Europe, climate change has created favorable microenvironments for mosquitoes and ticks in areas previously considered at low-risk”
Thank you, the paragraph has been revised and adapted with updated bibliographic sources.
Table 1.
Please correct the sentence “Pathologies without autochthonous transmission in humans or animals and without the presence of competent vectors in Europe” as “Pathogens without autochthonous transmission in humans or animals and without the presence of competent vectors in Europe”
Thank you, we have modified the table inserting the disease name.
It is good if authors provide information how climate change, globalization, wild life and ecological disruptions increase the introduction, distribution and transmission of arthropod-borne arbovirus in Europe.
Thank you for the comment. The topic has been expanded with explanations regarding the interactions among the various determinants of VBDs, specifically focusing on the interactions between the environment, humans, and disease.
CONCLUSIONS
This section seems good
References
Authors should avoid citing articles published in predatory journals and replace them with references from high-quality international journals
Thank you, we have revised all the references try to insert articles from high-quality international journals
Reviewer 5 Report
Comments and Suggestions for Authors
This is a nice and comprehensive review of the vector-borne infectious diseases which, because of the climate change, are increasingly affecting Europe. The review is well-written, and the data reported are updated.
Suggestions:
1. An area that can be improved is the link between the vectors and the outbreaks reported in Europe. It is difficult to link the vector distributions in table 1 and Figure 1 to the cases reported in Europe. For instance, it is not clear which vector caused the cases of dengue in Italy or France.
2. I suggest adding one figure describing the viral outbreaks in Europe reported as circles with the diameter proportional to the number of cases.
Author Response
This is a nice and comprehensive review of the vector-borne infectious diseases which, because of the climate change, are increasingly affecting Europe. The review is well-written, and the data reported are updated.
Suggestions:
- An area that can be improved is the link between the vectors and the outbreaks reported in Europe. It is difficult to link the vector distributions in table 1 and Figure 1 to the cases reported in Europe. For instance, it is not clear which vector caused the cases of dengue in Italy or France.
Thank you for the valuable suggestion, however we hope that the description made in the text can make up for the lack of greater clarity in figure 1 and in the text. It has been reported in the Dengue virus section, in fact, that Aedes aegypti, the main vector responsible for the Dengue virus, is present in Europe only in small areas, while Aedes albopictus is widely present as shown in figure 1a and figure 3
- 2. I suggest adding one figure describing the viral outbreaks in Europe reported as circles with the diameter proportional to the number of cases.
Thanks for the constructive suggestion, but given the large number of outbreaks for each virus described, we had difficulty setting up a figure as required. We trust that this will not be judged negatively by the referee and will not compromise the general opinion on the work
Round 2
Reviewer 1 Report
Comments and Suggestions for Authors
Review – version 2
General comments:
The article has shown a great improvement after the first round of review. I appreciate the modifications made throughout the manuscript, and I recommend publishing the article.
Please follow below some remaining minor points to re-check and address in the final version, especially in the Conclusion section.
Abstract:
General comment: I appreciate the improvement in the abstract section.
Introduction:
General comment: Overall, I enjoyed reading the new version introduction. The authors included all the information necessary to provide an expensive background about the arboviruses' situation, as well as the features associated with its increase and spreading. However, for the final version, I suggest checking the position of Table 1 and Table 1 heading, which is out of place (lines 82-83) and some random letters on lines 142 to 148.
Session 2:
General Comment: The search and inclusion/exclusion criteria are clear now.
Specific viruses sections:
General Comment: Reading is more fluid, and the cohesion of ideas has been dramatically improved.
Lines 411-412: Provide the complete reference for the Tick factsheets (adapted from …, available on …, accessed MM/DD/YYYY).
Conclusion:
General Comment: Thank authors for addressing all the points, however, I couldn’t find the reference citations after line 670. If there is no article, I suggest searching for articles on the same topic applied to other countries or regions across the globe.
Author Response
General comments:
The article has shown a great improvement after the first round of review. I appreciate the modifications made throughout the manuscript, and I recommend publishing the article.
Please follow below some remaining minor points to re-check and address in the final version, especially in the Conclusion section.
Abstract:
General comment: I appreciate the improvement in the abstract section.
Thank you
Introduction:
General comment: Overall, I enjoyed reading the new version introduction. The authors included all the information necessary to provide an expensive background about the arboviruses' situation, as well as the features associated with its increase and spreading. However, for the final version, I suggest checking the position of Table 1 and Table 1 heading, which is out of place (lines 82-83) and some random letters on lines 142 to 148.
Thank you, we have corrected the position of the Table 1 and have deleted the letters
Session 2:
General Comment: The search and inclusion/exclusion criteria are clear now.
Thank you
Specific viruses sections:
General Comment: Reading is more fluid, and the cohesion of ideas has been dramatically improved.
Thank you
Lines 411-412: Provide the complete reference for the Tick factsheets (adapted from …, available on …, accessed MM/DD/YYYY).
We have added the complete reference
Conclusion:
General Comment: Thank authors for addressing all the points, however, I couldn’t find the reference citations after line 670. If there is no article, I suggest searching for articles on the same topic applied to other countries or regions across the globe.
We have added the references
Reviewer 2 Report
Comments and Suggestions for Authors
Revision needed
West Nile virus and Usutu virus need to be included
Author Response
Revision needed
West Nile virus and Usutu virus need to be included
We regret not having satisfied this requirement of the reviewer, but as explained in the selection criteria section, we have excluded Usutu virus, West Nile virus and tick-borne encephalitis because they are already considered endemic-epidemic in Europe (https://doi.org/10.1016/j.onehlt.2023.100664 ; https://doi.org/10.2807/ese.16.31.19935-en) and to treat also these viruses with the many outbreaks due to them would have further lengthened this review. We think that all these viruses and the related vectors and diseases deserve a separate discussion in another paper
Reviewer 3 Report
Comments and Suggestions for Authors
Thank you foe addressing the concerns raised
Author Response
Comments and Suggestions for Authors
Thank you foe addressing the concerns raised
Thank you
Round 3
Reviewer 2 Report
Comments and Suggestions for Authors
Revision not completed
Author Response
Revision not completed
We have added a brief sentence at the end of the Introduction section underlining the reason by which West Nile and Usutu viruses have not been included